# Olfactory connectivity mediates sleep-dependent food choices in humans

**Surabhi Bhutani[1,2], James D Howard[1], Rachel Reynolds[1], Phyllis C Zee[1], Jay Gottfried[1,3,4], Thorsten Kahnt[1,5,6]***

[1]Department of Neurology, Feinberg School of Medicine, Northwestern University, Chicago, United States; [2]School of Exercise and Nutritional Sciences, College of Health and Human Services, San Diego State University, San Diego, United States; [3]Department of Neurology, Perelman School of Medicine, University of Pennsylvania, Philadelphia, United States; [4]Department of Psychology, School of Arts and Sciences, University of Pennsylvania, Philadelphia, United States; [5]Department of Psychiatry and Behavioral Sciences, Feinberg School of Medicine, Northwestern University, Chicago, United States; [6]Department of Psychology, Weinberg College of Arts and Sciences, Northwestern University, Evanston, United States

**Abstract** Sleep deprivation has marked effects on food intake, shifting food choices toward energy-dense options. Here we test the hypothesis that neural processing in central olfactory circuits, in tandem with the endocannabinoid system (ECS), plays a key role in mediating this relationship. We combined a partial sleep-deprivation protocol, pattern-based olfactory neuroimaging, and *ad libitum* food intake to test how central olfactory mechanisms alter food intake after sleep deprivation. We found that sleep restriction increased levels of the ECS compound 2-oleoylglycerol (2-OG), enhanced encoding of food odors in piriform cortex, and shifted food choices toward energy-dense food items. Importantly, the relationship between changes in 2-OG and food choices was formally mediated by odor-evoked connectivity between the piriform cortex and insula, a region involved in integrating feeding-related signals. These findings describe a potential neurobiological pathway by which state-dependent changes in the ECS may modulate chemosensory processing to regulate food choices.
DOI: https://doi.org/10.7554/eLife.49053.001

*For correspondence:
thorsten.kahnt@northwestern.edu

## Introduction

Sleep deprivation profoundly impacts food choices. When individuals are sleep-deprived, their dietary behavior shifts toward increased consumption of foods high in sugar and fat, leading to weight gain (*Markwald et al., 2013*; *Nedeltcheva et al., 2009*). These effects on ingestive behavior are likely related to sleep-dependent changes in appetite-regulating compounds, including ghrelin (*Rihm et al., 2019*; *Spiegel et al., 2004b*), leptin (*Spiegel et al., 2004a*), and endocannabinoids (*Hanlon et al., 2016*). Indeed, the endocannabinoid system (ECS) exerts strong effects on food intake (*Bellocchio et al., 2010*; *Di Marzo et al., 2001*), and levels of the endocannabinoid 2-arachidonoylglycerol (2-AG) and its structural analog 2-oleoylglycerol (2-OG) are enhanced in sleep-deprived individuals (*Hanlon et al., 2016*). While previous studies have tested the effects of sleep deprivation on the human brain (*Greer et al., 2013*; *Krause et al., 2017*; *Muto et al., 2016*; *Rihm et al., 2019*), the neural pathways through which sleep-dependent alterations in the ECS influence food intake have not been investigated in humans.

**eLife digest** People who do not get enough sleep often start to favor sweet and fatty foods, which contributes to weight gain. While the exact mechanisms are still unknown, lack of sleep seems to change food preferences by influencing the levels of molecules that regulate food intake. In particular, it could have an effect on the endocannabinoid system, a complex network of molecules in the nervous system that controls biological processes such as appetite.

The sense of smell is also tightly linked to how and what organisms choose to eat. Recent experiments indicate that in rodents, endocannabinoids enhance food intake by influencing the activity of the brain areas that process odors. However, it is still unclear whether the brain regions that process odors play a similar role in humans.

To investigate, Bhutani et al. examined the impact of a four-hour night's sleep on 25 healthy human volunteers. Blood analyses showed that after a short night, individuals had increased amounts of 2-oleoylglycerol, a molecule that is part of the endocannabinoid system. When sleep-deprived people were given the choice to eat whatever they wanted, those with greater levels of 2-oleoylglycerol preferred food higher in energy. Bhutani et al. also imaged the volunteers' brains to examine whether these changes were connected to modifications in the way the brain processed smells. This revealed that, in people who did not sleep enough, an odor-processing region called the piriform cortex was encoding smells more strongly.

The piriform cortex is connected to another region, the insula, which integrates information about the state of the body to control food intake. Lack of sleep altered this connection, and this was associated with a preference for high-energy food. In addition, further analysis showed that changes in the amounts of 2-oleoylglycerol were linked to modifications in the connection between the two brain areas. Taken together, these results suggest that sleep deprivation influences the endocannabinoid system, which in turn alters the connection between piriform and insular cortex, leading to a shift toward foods which are high in calories.

In the United States alone, one in three people sleep less than six hours a night. Learning more about how sleep deprivation affects brain pathways and food choice may help scientists to develop new drugs or behavioral therapies for conditions like obesity.

DOI: https://doi.org/10.7554/eLife.49053.002

One likely target for sleep-dependent neuromodulation of food intake is the olfactory system. Odors serve as powerful signals for the initiation and termination of feeding behavior (*Saper et al., 2002*; *Shepherd, 2006*), and animal studies have shown that olfactory processing is modulated in a state-dependent manner (*Julliard et al., 2007*; *McIntyre et al., 2017*; *Murakami et al., 2005*). In rodents (*Aimé et al., 2007*; *Aimé et al., 2014*) and humans (*Hanci and Altun, 2016*; *Stafford and Welbeck, 2011*), olfaction is altered by hunger and satiety, and satiety reduces neural activity in olfactory brain regions in parallel with a suppression of feeding behavior (*Boesveldt, 2017*; *Gervais and Pager, 1979*; *O'Doherty et al., 2000*; *Prud'homme et al., 2009*; *Soria-Gómez et al., 2014*). Moreover, recent work across different species suggests a link between the ECS, olfactory processing, and food intake, such that endocannabinoids may directly modulate neural activity in olfactory circuits (*Breunig et al., 2010*; *Soria-Gómez et al., 2014*). However, whether odor-evoked responses in the human olfactory system are similarly modulated by the ECS, and whether this accounts for the effects of sleep deprivation on food intake, is not known.

We hypothesized that sleep deprivation is associated with a cascade of metabolic and olfactory changes, ultimately steering food choices toward energy-dense options (*Simon et al., 2015*). We predicted that after a night of restricted sleep, relative levels of circulating ECS compounds will be increased (*Hanlon et al., 2016*), leading to changes in how olfactory brain regions in the medial temporal and basal frontal lobes respond to food odors (*Soria-Gómez et al., 2014*). We expected that such sleep-dependent changes in olfactory processing would manifest in odor-evoked activity patterns in piriform cortex (*Howard and Gottfried, 2014*; *Howard et al., 2009*), and that effects on food intake would involve interactions with areas downstream of piriform cortex, such as the insula. Olfactory, gustatory, homeostatic, and visceral signals are integrated in the insula (*Craig, 2002*; *de Araujo et al., 2003*; *Johnson et al., 2000*; *Livneh et al., 2017*; *Small et al., 2008*), optimally

positioning this region to regulate ingestive behavior in a state-dependent manner (*Dagher, 2012*; *de Araujo et al., 2006*).

## Results

### Experimental design and sleep deprivation

To test these hypotheses, we utilized a within-subject crossover design with a partial sleep-deprivation protocol and pattern-based functional magnetic resonance imaging (fMRI) of food and non-food odors (*Figure 1A*). The experiment was designed to simultaneously measure the effects of sleep deprivation on ECS signaling, neural responses to food odors, and real-life food choices. After one week of sleep stabilization (7–9 hr sleep/night between 10:30 pm and 7:30 am), healthy-weight participants (N = 25, 10 male, age mean ± SEM: 26.6 ± 0.98 years) were randomly assigned to one night of deprived sleep (DS, 4 hr sleep between 1 am and 5 am) or non-deprived sleep (NDS, 8 hr sleep between 11 pm and 7 am). All subjects participated in both DS and NDS sessions, which were separated by 4 weeks to allow for sufficient recovery time and to control for potential effects of menstrual phase in female participants. Actigraphy-monitored sleep times did not differ between DS and NDS sessions during the 7-nights of sleep stabilization (NDS: 6.63 ± 0.18 hr, DS: 6.75 ± 0.19 hr; $T_{22}=-1.40$, p=0.174). However, sleep times did differ during the night of sleep manipulation (NDS: 6.8 ± 0.12 hr, DS: 3.8 ± 0.18 hr; $T_{24} = 14.70$, p=$1.68 \times 10^{-13}$, *Figure 1B*, *Figure 1—figure supplement 1*, and *Supplementary file 1*), confirming that subjects complied with the sleep deprivation protocol.

The functional imaging sessions occurred in the evening after the night of sleep manipulation. In the 24 hr period leading up to these critical testing sessions, subjects received individually standardized isocaloric diets to ensure identical food intake in both sessions. Subjective states of sleep deprivation were assessed in the morning, and standardized sleepiness scores (Stanford sleepiness scale) were obtained upon arrival at the imaging center. Compared to the NDS session, participants in the DS session felt subjectively sleep-deprived, as indicated by reduced self-reported sleep quality, levels of alertness and well-restedness (*Figure 1D*), and increased sleepiness (*Figure 1E*). To account for potential effects of sleep deprivation-related stress and anxiety on food intake and associated brain responses (*Maier et al., 2015*), subjects completed the State Anxiety Inventory (SAI). We also measured serum cortisol levels as a physiological marker of stress. There were no significant changes in SAI scores ($T_{24}=-1.49$, p=0.148) or cortisol ($T_{24} = 0.70$, p=0.584) between DS and NDS sessions (*Figure 1—figure supplement 2*), suggesting that stress and anxiety levels were not altered by sleep restriction.

Before fMRI scanning, subjects received a standardized isocaloric dinner based on individually estimated energy demands. Hunger ratings decreased immediately after dinner, and returned to pre-dinner levels 45 min after the meal (*Figure 1C*). Importantly, there were no sleep-dependent differences in hunger ratings made at any time point, indicating that measures of odor-evoked brain activity and food intake cannot be driven by differences in subjective levels of hunger. During fMRI scanning, subjects intermittently smelled a set of energy-dense food odors and non-food control odors (*Supplementary file 2*). There were no sleep-dependent differences in rated odor pleasantness and intensity, or respiratory behavior during fMRI scanning (*Figure 1—figure supplement 3*).

### Sleep deprivation shifts food choices toward energy-dense options

Immediately after scanning, subjects were given *ad libitum* access to food items in a buffet-style setting to measure effects of sleep deprivation on food choices and intake. In the DS session, participants consumed food items with a significantly higher energy density (6.0 ± 2.43% change from NDS; $T_{24} = 2.48$, p=0.021, *Figure 1F*), with no significant difference in the overall calories consumed (18.6 ± 16.93% change from NDS; $T_{24} = 1.10$, p=0.282). Importantly, effects of sleep deprivation on dietary behavior persisted into the next day (after a night of unrestricted recovery sleep), with a higher percentage of calories consumed as fat (DS: 36.5 ± 1.28%, NDS: 30.6 ± 1.28%; $T_{24} = 2.34$, p=0.028), indicating that a single night of restricted sleep can have relatively long-lasting effects on food choices.

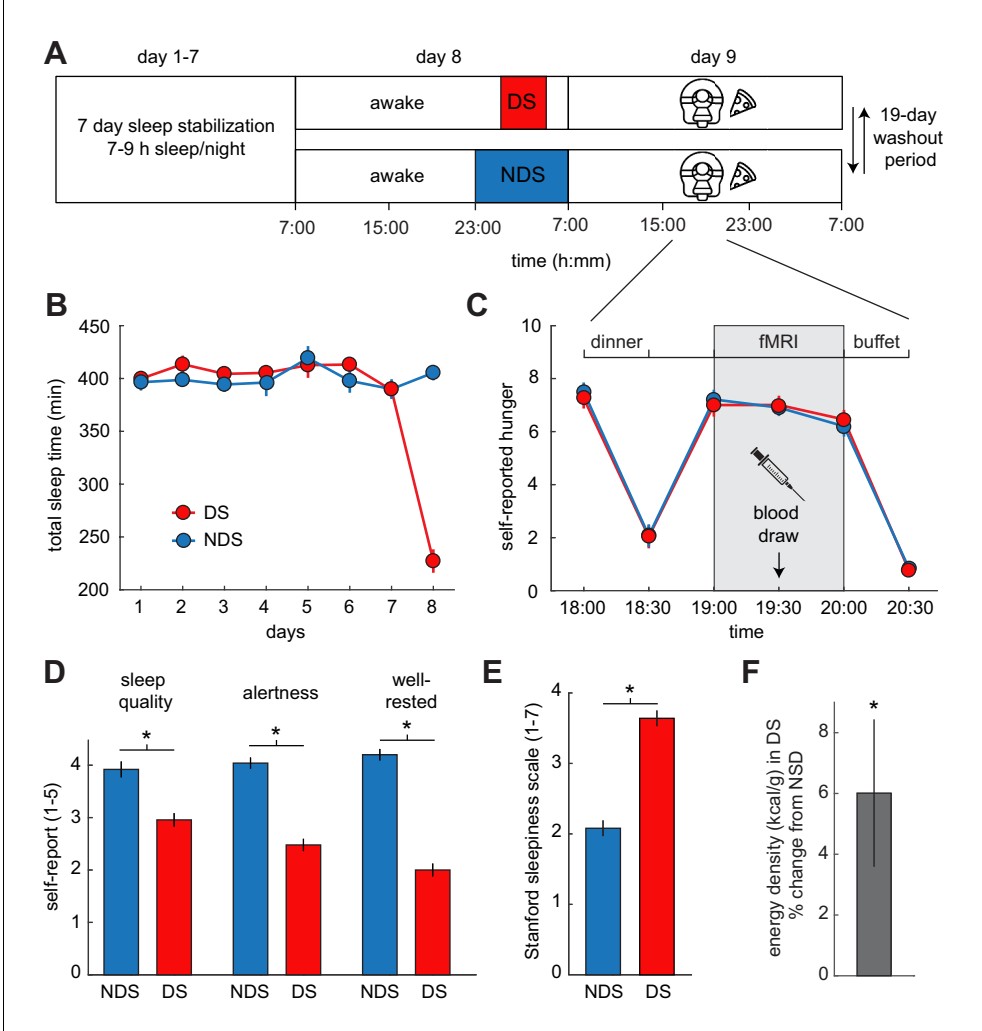

**Figure 1.** Experimental design and behavioral effects of sleep deprivation. (**A**) Study protocol for deprived sleep (DS) and non-deprived sleep (NDS) sessions with a 19 day washout period. Dinner was served at 6 pm, the fMRI session started at 7 pm, and the *ad libitum* buffet started after 8 pm. (**B**) Actigraphy data showed no differences in sleep duration during the sleep stabilization phase, but confirmed a significant difference between the two sleep conditions during the night of sleep manipulation (day 8). (**C**) Ratings of hunger on a visual analog scale showed significant effects of time, but no sleep-dependent effects (time-by-sleep ANOVA, main effect time, $F_{5,120}=70.86$, $p=3.63\times10^{-34}$, main effect sleep, $F_{1,24}=0.06$, $p=0.81$, interaction $F_{5,120}=0.24$, $p=0.95$). (**D**) In the DS compared to the NDS session, participants reported lower sleep quality (DS $2.96 \pm 0.19$, NDS $3.96 \pm 0.18$, $T_{22}=-3.98$, $p=6.38\times10^{-4}$), reduced alertness (DS $2.49 \pm 0.14$, NDS $4.09 \pm 0.14$, $T_{22}=-8.66$, $p=1.55\times10^{-8}$), and felt less well-rested (DS $2.00 \pm 0.14$, NDS $4.17 \pm 0.17$, $T_{22}=-9.72$, $p=2.01\times10^{-9}$). (**E**) Stanford Sleepiness Scale scores were higher in the DS session (DS $3.64 \pm 0.24$, NDS $2.08 \pm 0.19$, $T_{24} = 6.96$, $p=3.40\times10^{-7}$). (**F**) Energy-density (kcal/g) of food consumed after scanning at the *ad libitum* buffet, expressed as % change from NDS baseline. *p<0.05. Data are presented as mean ± SEM.

DOI: https://doi.org/10.7554/eLife.49053.003

The following source data and figure supplements are available for figure 1:

**Source data 1.** Relates to panel (**B**).
DOI: https://doi.org/10.7554/eLife.49053.008
**Source data 2.** Relates to panel (**C**).
DOI: https://doi.org/10.7554/eLife.49053.009
**Source data 3.** Relates to panel (**D**).
DOI: https://doi.org/10.7554/eLife.49053.010
**Source data 4.** Relates to panel (**E**).
DOI: https://doi.org/10.7554/eLife.49053.011

*Figure 1 continued on next page*

*Figure 1 continued*

**Source data 5.** Relates to panel (F).
DOI: https://doi.org/10.7554/eLife.49053.012

**Figure supplement 1.** Actigraphy data.
DOI: https://doi.org/10.7554/eLife.49053.004

**Figure supplement 1—source data 1.** Relates to panel (A).
DOI: https://doi.org/10.7554/eLife.49053.013

**Figure supplement 1—source data 2.** Relates to panel (B).
DOI: https://doi.org/10.7554/eLife.49053.014

**Figure supplement 1—source data 3.** Relates to panel (C).
DOI: https://doi.org/10.7554/eLife.49053.015

**Figure supplement 1—source data 4.** Relates to panel (D).
DOI: https://doi.org/10.7554/eLife.49053.016

**Figure supplement 1—source data 5.** Relates to panel (E).
DOI: https://doi.org/10.7554/eLife.49053.017

**Figure supplement 1—source data 6.** Relates to panel (F).
DOI: https://doi.org/10.7554/eLife.49053.018

**Figure supplement 2.** No effects of sleep deprivation on anxiety and hormones.
DOI: https://doi.org/10.7554/eLife.49053.005

**Figure supplement 2—source data 7.** Relates to panel (A).
DOI: https://doi.org/10.7554/eLife.49053.019

**Figure supplement 2—source data 8.** Relates to panel (B).
DOI: https://doi.org/10.7554/eLife.49053.020

**Figure supplement 3.** Odor pleasantness and intensity ratings, and respiratory responses during fMRI.
DOI: https://doi.org/10.7554/eLife.49053.006

**Figure supplement 3—source data 9.** Relates to panel (A).
DOI: https://doi.org/10.7554/eLife.49053.021

**Figure supplement 3—source data 10.** Relates to panel (B).
DOI: https://doi.org/10.7554/eLife.49053.022

**Figure supplement 3—source data 11.** Relates to panel (C).
DOI: https://doi.org/10.7554/eLife.49053.023

**Figure supplement 3—source data 12.** Relates to panel (D).
DOI: https://doi.org/10.7554/eLife.49053.024

**Figure supplement 3—source data 13.** Relates to panel (E).
DOI: https://doi.org/10.7554/eLife.49053.025

**Figure supplement 3—source data 14.** Relates to panel (F).
DOI: https://doi.org/10.7554/eLife.49053.026

**Figure supplement 4.** Individual data points for sleep quality, sleepiness, and energy-dense food choices.
DOI: https://doi.org/10.7554/eLife.49053.007

**Figure supplement 4—source data 15.** Relates to panel (A).
DOI: https://doi.org/10.7554/eLife.49053.027

**Figure supplement 4—source data 16.** Relates to panel (B).
DOI: https://doi.org/10.7554/eLife.49053.028

**Figure supplement 4—source data 17.** Relates to panel (C).
DOI: https://doi.org/10.7554/eLife.49053.029

## Sleep-dependent changes in the ECS correlate with energy-dense food choices

Based on previous reports that circulating levels of 2-AG and 2-OG are enhanced after sleep restriction (*Hanlon et al., 2016*), we next tested whether these ECS compounds were increased in the present study. Partially replicating the previous findings, circulating levels of 2-OG collected during fMRI scanning were increased in the DS relative to the NDS session ($35.02 \pm 17.82\%$ change from NDS; $T_{24} = 1.96$, p=0.031, one-tailed, *Figure 2A*). However, although sleep-dependent changes in 2-AG and 2-OG were significantly correlated (r = 0.55, p=0.004), relative increases in 2-AG were not significant ($5.29 \pm 5.20\%$ change from NDS; $T_{24} = 1.02$, p=0.16, one-tailed). We also observed no

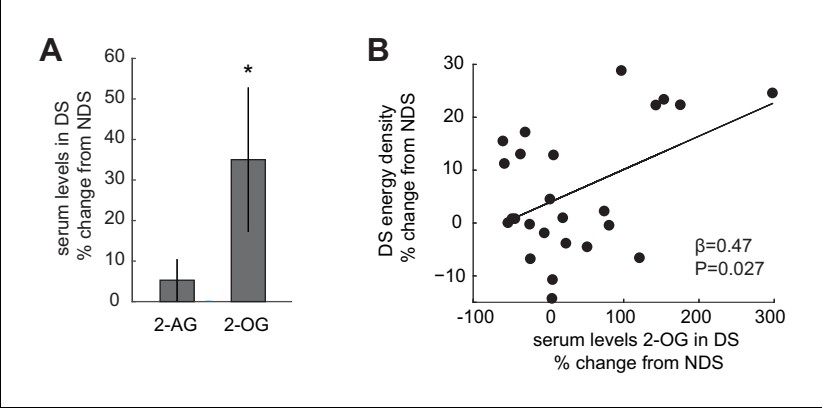

**Figure 2.** Sleep-dependent changes in the ECS correlate with energy-dense food choices. (**A**) Relative changes in 2-AG and 2-OG levels in the SD condition (% change from NDS baseline). Data are presented as mean ± SEM. (**B**) Percent changes in 2-OG in DS from NDS baseline correlate positively with % changes in energy density of food consumed at the *ad libitum* buffet in the DS condition relative to NDS baseline (robust regression, β = 0.47, p=0.027).
DOI: https://doi.org/10.7554/eLife.49053.030

The following source data and figure supplements are available for figure 2:

**Source data 1.** Relates to panel (**A**).
DOI: https://doi.org/10.7554/eLife.49053.032
**Source data 2.** Relates to panel (**B**).
DOI: https://doi.org/10.7554/eLife.49053.033
**Figure supplement 1.** Individual data points for 2-AG and 2-OG.
DOI: https://doi.org/10.7554/eLife.49053.031
**Figure supplement 1—source data 1.** Relates to panel (**A**).
DOI: https://doi.org/10.7554/eLife.49053.034
**Figure supplement 1—source data 2.** Relates to panel (**B**).
DOI: https://doi.org/10.7554/eLife.49053.035

significant changes in other appetite-regulating hormones, including ghrelin, leptin, and insulin (*Figure 1—figure supplement 2B*). Interestingly, sleep-dependent increases in 2-OG correlated significantly with increases in the energy density of food consumed at the post-scanning buffet (robust regression, β = 0.47, p=0.027; permutation test, p=0.018; *Figure 2B*). Although correlative in nature, this finding suggests that the ECS may play a role in modifying dietary behavior after sleep deprivation.

## Sleep deprivation enhances odor encoding in piriform cortex

Having established a link between sleep-dependent changes in the ECS and food choices, we next analyzed the fMRI data to examine whether this relationship was mediated by effects of sleep deprivation on central olfactory responses to odors. Based on previous rodent work showing that endocannabinoids affect feeding-related changes in olfactory processing (*Soria-Gómez et al., 2014*), we hypothesized that elevated levels of 2-OG would be accompanied by enhanced representations of odors in olfactory sensory cortices.

Both animal (*Barnes et al., 2008*; *Illig and Haberly, 2003*; *Stettler and Axel, 2009*) and human studies (*Howard et al., 2009*; *Zelano et al., 2011*) have shown that odors are encoded in piriform cortex by sparsely distributed patterns of ensemble activity, with no apparent topographical organization. Such distributed representations cannot be detected by univariate fMRI analyses, in which activity is typically averaged across voxels, thus blurring information contained within fine-grained patterns of activity. To examine such distributed responses to food odors (*Figure 3A*), here we used a searchlight-based multi-voxel pattern analysis (MVPA), which enables unbiased whole-brain decoding based on activity patterns (*Haynes et al., 2007*; *Kahnt et al., 2011*; *Kriegeskorte et al., 2006*).

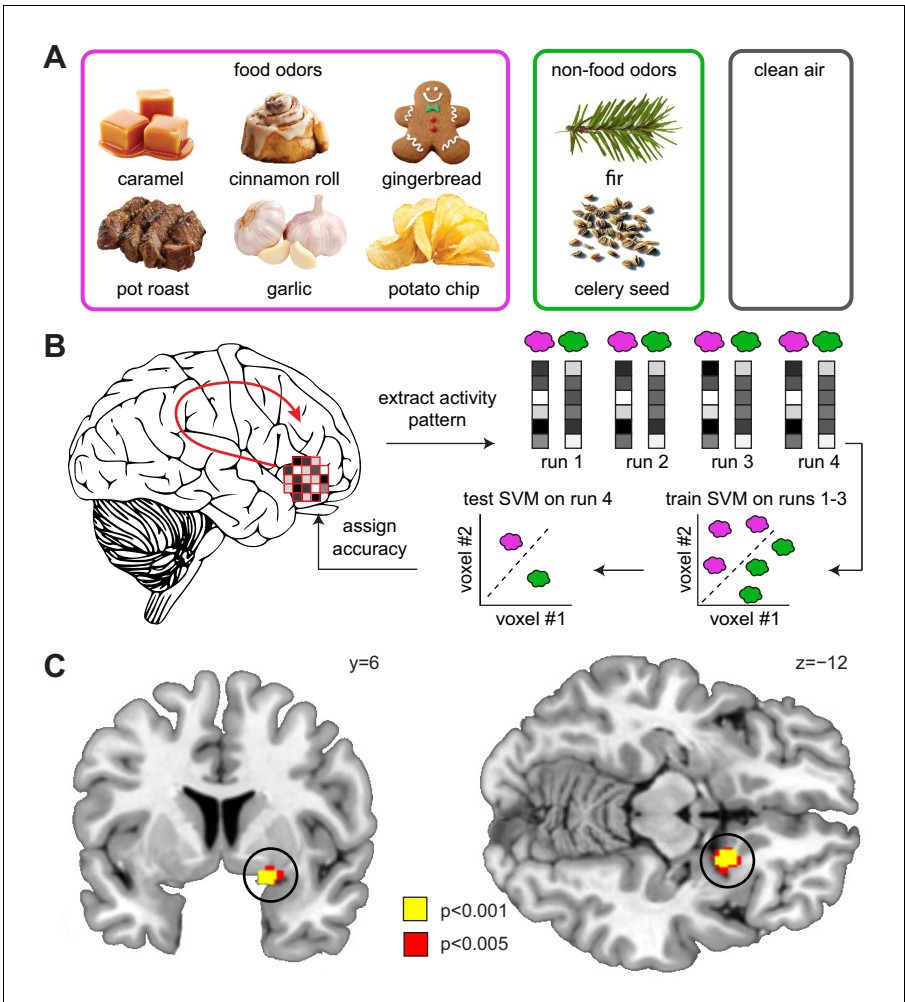

**Figure 3.** Sleep deprivation enhances encoding of odor information in piriform cortex. (**A**) Sweet and savory food odors and non-food control odors presented during fMRI. (**B**) Schematic of the searchlight decoding analysis used to reveal information about food vs. non-food odors in the DS compared to the NDS session. (**C**) Decoding accuracy for food vs. non-food odors in the piriform cortex (black circle) was significantly higher in the DS compared to the NDS session ($T_{24}$ = 5.91, $P_{FWE-SVC}$ = 0.001). This result did not change when including covariates for head motion (translation and rotation) into the group-level model (x = 20 y = 8 z=−12, $T_{23}$ = 6.12, $P_{FWE-SVC}$ = 0.0001). In addition, results were still significant when including covariates for odor pleasantness in the first- (x = 20, y = 8, z=−10, $T_{24}$ = 3.12, $P_{FWE-SVC}$ = 0.045) and group-level models (x = 20, y = 8, z=−12, $T_{23}$ = 6.34, $P_{FWE-SVC}$ = 0.0001). Finally, controlling for respiratory response functions (***Birn et al., 2008***) did not change the result (x = 20, y = 8, z=−12, $T_{24}$ = 4.83, $P_{FWE-SVC}$ = 0.001). Whole-brain map can be viewed at neurovault.org/images/132917/.

DOI: https://doi.org/10.7554/eLife.49053.036

The following figure supplements are available for figure 3:

**Figure supplement 1.** Decoding of odor information across sleep sessions.

DOI: https://doi.org/10.7554/eLife.49053.037

**Figure supplement 2.** fMRI signal coverage.

DOI: https://doi.org/10.7554/eLife.49053.038

Specifically, we used a support vector machine (SVM) classifier to decode information about food vs. non-food odors from patterns of odor-evoked fMRI activity (***Figure 3B***). Across sleep sessions, we found significant decoding of odor information in the piriform cortex (x = 16, y=−2, z=−14, $T_{24}$ = 4.20, $P_{FWE-SVC}$ = 0.012, ***Figure 3—figure supplement 1***) and insula (left x=−32, y=−4, z = 16, $T_{24}$ = 4.75, $P_{FWE-SVC}$ = 0.021; right x = 44, y = 8, z = 10, $T_{24}$ = 4.84, $P_{FWE-SVC}$ = 0.017). Importantly,

comparing odor encoding between DS and NDS sessions, we found significantly higher decoding accuracy in the piriform cortex in the DS session (x = 20, y = 8, z=−12, $T_{24}$ = 5.91, $P_{FWE-SVC}$ = 0.001; *Figure 3C*), suggesting that sleep deprivation enhances encoding of odor information in olfactory brain areas. In contrast, a univariate analysis of odor-evoked fMRI responses in piriform cortex showed no sleep-dependent effects or interactions (*Figure 4*).

Enhanced encoding of odor information after sleep deprivation could be the mediating neural factor behind the observed relationship between sleep-dependent changes in the ECS and food choices. However, decoding accuracy in piriform cortex was not correlated with either sleep-dependent changes in 2-OG (r=−0.24, p=0.248) or energy-dense food choices (r=−0.05, p=0.80), suggesting that the relationship between the ECS and food intake was not directly mediated by changes in odor encoding. In the next step, we therefore considered the possibility that changes in the propagation of olfactory signals from piriform cortex to downstream regions may account for the effects of sleep deprivation on food choices.

## Piriform-insula connectivity mediates the link between ECS and food intake

One downstream region particularly relevant for integrating chemosensory, interoceptive, and homeostatic signals to guide food intake is the insula (*Dagher, 2012*; *de Araujo et al., 2006*; *Livneh et al., 2017*). To test whether sleep deprivation altered the functional connectivity between piriform and insular cortex, we utilized a psychophysiological interaction (PPI) model, with piriform cortex as the seed region and odor presentation (food and non-food odors > clean air) as the psychological variable. We found that sleep-dependent changes (DS >NDS) in piriform connectivity (odorized >clean air) with the right insula correlated with changes in the energy density of food consumed immediately after the fMRI session (x = 40, y = 6, z = 0, $T_{23}$ = 6.04, $P_{FWE-SVC}$ = 0.005; *Figure 5A*), such that reduced odor-evoked connectivity was associated with enhanced intake of energy-dense food (*Figure 5B*). A similar effect in the left insula was present but did not survive correction for multiple comparisons (x=−44, y = 4, z = 0, $T_{23}$ = 4.20, $P_{FWE-SVC}$ = 0.17). In addition, while not part of our main set of hypotheses, we found a positive correlation between sleep-dependent changes in energy-dense food choices and changes in odor-evoked connectivity between the piriform cortex and the right anterior hippocampus (x = 28, y=−10, z=−18, $T_{23}$ = 5.36,

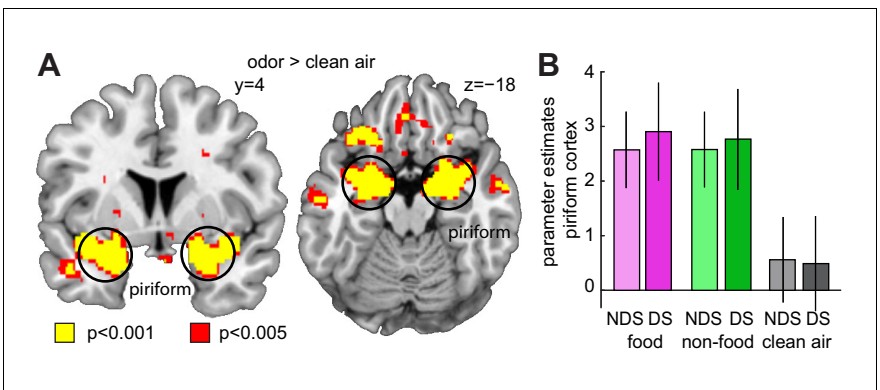

**Figure 4.** Sleep deprivation does not enhance univariate fMRI responses to odors. (A) Significant univariate odor-evoked fMRI responses (food and non-food odors > clean air) in piriform cortex, averaged across both sleep sessions (right, x = 22, y = −4, z = −18, $T_{24}$ = 12.53, $P_{FWE}$ = 4.42×$10^{-7}$; left, x = −24, y = 4, z = −18, $T_{24}$ = 9.06, $P_{FWE}$ = 2.83×$10^{-4}$). Whole-brain map can be viewed at neurovault.org/images/132916/ (B) Parameter estimates in piriform cortex show no differences between food and non-food odors, and no sleep-dependent effects (two-way ANOVA, main effect of sleep, $F_{1,24}$=0.003, p=0.954; main effect of odor, $F_{2,48}$=112.88, p=7.12×$10^{-19}$; sleep-by-odor interaction, $F_{2,482}$=0.29, p=0.748). Data are represented as mean ± SEM.
DOI: https://doi.org/10.7554/eLife.49053.039

The following source data is available for figure 4:

**Source data 1.** Relates to panel (B).
DOI: https://doi.org/10.7554/eLife.49053.040

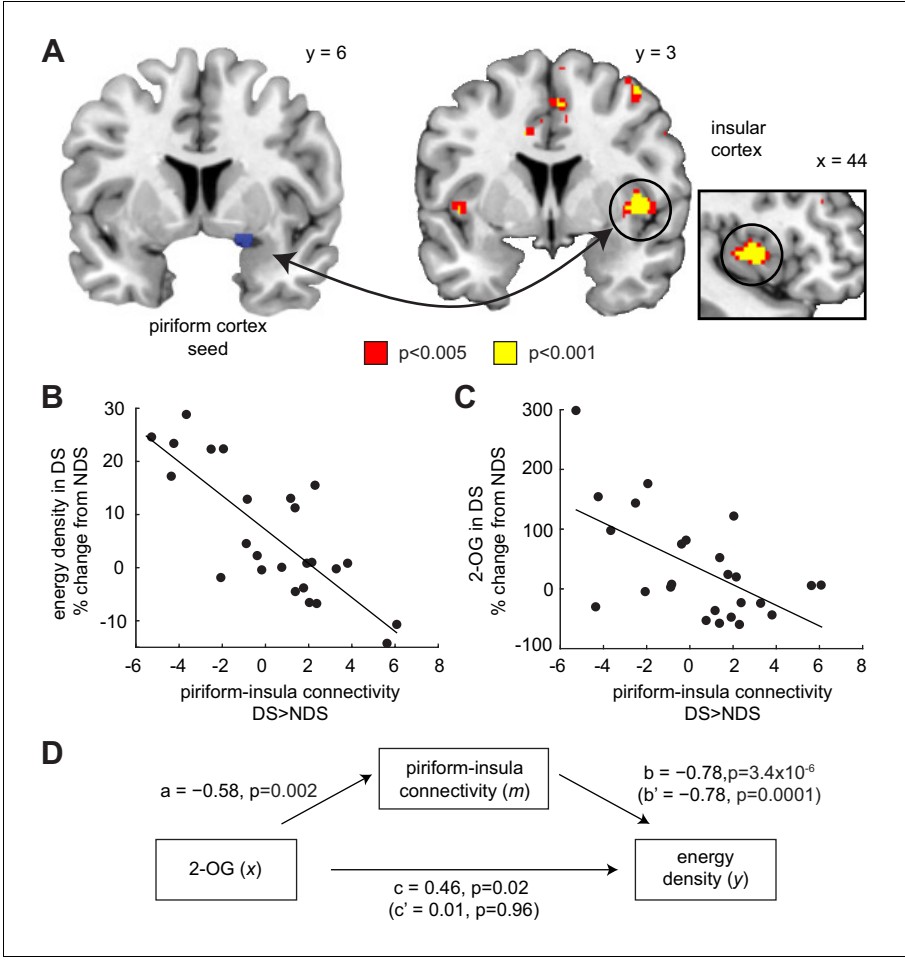

**Figure 5.** Piriform-insula connectivity mediates the effects of 2-OG on sleep-dependent food choices. (**A**) Sleep-dependent changes in odor-evoked connectivity (odor >clean air) between piriform cortex and insula negatively correlated with the energy density of foods consumed at the post-fMRI buffet ($T_{23}$ = 6.04, $P_{FWE-SVC}$ = 0.005). This result did not change when including covariates for head motion (translation and rotation) into the group-level model (x = 40, y = 6, z = 0, $T_{21}$ = 5.42, $P_{FWE-SVC}$ = 0.021). In addition, controlling for respiratory response functions (*Birn et al., 2008*) did not change the result (x = 40, y = 6, z = 0, $T_{23}$ = 5.59, $P_{FWE-SVC}$ = 0.011). Whole-brain map can be viewed at neurovault.org/images/132919/ (**B and C**) For illustrative purposes, scatter plots depict association between sleep-dependent changes in piriform-insula connectivity and (**B**) energy density of food intake (r=−0.78, p=3.4×10$^{-6}$) and (**C**) 2-OG (r=−0.58, p=0.002). (**D**) Mediation analysis. Sleep-dependent changes in 2-OG separately correlated with choices of energy-dense foods (c = 0.46, p=0.019) and piriform-insular connectivity (a=−0.58, p=0.002), and piriform-insula connectivity correlated with food intake (b = 0.785, p=3.39×10$^{-6}$). The association between 2-OG and food choices was no longer significant when the indirect effect of piriform-insula connectivity on food choice was included in the regression model (c'=0.01, p=0.956), which itself remained significant when controlling for 2-OG (b'=−0.78, p=0.0001).
DOI: https://doi.org/10.7554/eLife.49053.041

The following source data is available for figure 5:

**Source data 1.** Relates to panel (**B**).
DOI: https://doi.org/10.7554/eLife.49053.042
**Source data 2.** Relates to panel (**C**).
DOI: https://doi.org/10.7554/eLife.49053.043

$P_{uncorr}$ = 1.0×10$^{-5}$). In contrast, sleep-dependent changes in piriform connectivity for food vs. non-food odors did not show a significant relationship with changes in food choices. These results suggest that the connectivity between olfactory signals in the piriform cortex and downstream areas

may play a role in linking sleep-dependent changes in olfactory processing to changes in food choices.

Given that energy-dense food choices were associated with changes in 2-OG levels, it is possible that the observed connectivity effects were also related to the ECS. Indeed, we found that piriform-insula connectivity was significantly correlated with sleep-dependent changes in 2-OG levels (r=−0.58, p=0.002; *Figure 5C*), raising the possibility that the association between sleep-dependent changes in ECS and food choices is mediated by piriform-insula connectivity. To directly test this hypothesis, we employed a formal mediation analysis (*Baron and Kenny, 1986*), including a direct path from sleep-dependent changes in 2-OG (*x*) to changes in the energy density of the food consumed after fMRI (*y*), and an indirect path with changes in piriform-insula connectivity as mediator (*m*, *Figure 5D*). Importantly, the direct path between 2-OG levels and food intake was fully explained by the indirect path through piriform-insula connectivity, establishing a significant mediation effect (Sobel test, z = 2.97, p=0.003). This suggests that sleep deprivation affects the ECS, which then modulates the connectivity between piriform and insular cortex, and in turn shifts food choices toward energy-dense options.

## Discussion

Clinical and epidemiological studies have linked reduced sleep to elevated food intake and weight gain (*Kant and Graubard, 2014*; *Markwald et al., 2013*; *Patel and Hu, 2008*). This relationship has been confirmed in studies using experimentally induced sleep deprivation, demonstrating that sleep restriction increases the desire for foods high in sugar and fat content (*Cain et al., 2015*; *Greer et al., 2013*; *Hogenkamp et al., 2013*; *Simon et al., 2015*), and leads to excessive consumption of such food options (*Brondel et al., 2010*; *Nedeltcheva et al., 2009*). Several different factors have been proposed to account for this relationship (*Patel and Hu, 2008*), including changes in hunger induced by appetite-regulating hormones such as ghrelin and leptin (*Spiegel et al., 2004a*; *Spiegel et al., 2004b*), and the ECS (*Hanlon et al., 2016*). In addition, previous imaging studies have reported sleep-dependent activity changes in response to food cues (*Benedict et al., 2012*; *Greer et al., 2013*; *Rihm et al., 2019*; *St-Onge et al., 2014*), but whether and how these neural changes are related to actual food choices and the ECS has remained unclear.

In the current study, we tested the hypothesis that central olfactory mechanisms, in conjunction with the ECS, play a role in mediating the effects of sleep deprivation on dietary choices. This hypothesis was based on previous findings that experimentally induced sleep deprivation elevates relative levels of ECS compounds (*Hanlon et al., 2016*), and animal work suggesting that ECS activity drives changes in food intake through modulatory effects on olfactory circuits (*Breunig et al., 2010*; *Soria-Gómez et al., 2014*; *Wang et al., 2012*). In line with this idea, we found that sleep deprivation increased consumption of energy-dense foods, proportional to relative increases in 2-OG, and that it enhanced pattern-based encoding of odor information in the piriform cortex. Finally, we found that the effects of the ECS on food intake were mediated by changes in connectivity between the piriform cortex and the insula.

Previous animal studies have established an important role for the ECS in regulating feeding behavior (*Bellocchio et al., 2010*; *Di Marzo et al., 2001*; *Rodríguez de Fonseca et al., 2001*). More recently, it has been shown that levels of endocannabinoids in humans are elevated after sleep restriction (*Hanlon et al., 2016*), suggesting that this may drive altered dietary choices. However, unlike previous studies (*Hanlon et al., 2016*), we did not find a significant increase in 2-AG that paralleled increases in 2-OG. This may be due to the shorter duration of sleep restriction used in our study, but may also indicate different roles of the two compounds. Whereas 2-AG stimulates food intake and lipogenesis by activating CB1 receptors (*DiPatrizio and Simansky, 2008*; *Osei-Hyiaman et al., 2005*), the exact role of 2-OG and its relation to 2-AG is not fully understood (*Murataeva et al., 2016*). Our study shows that sleep-dependent increases in 2-OG are associated with changes in the consumption of energy-dense foods, providing novel evidence for a link between sleep, ECS, and dietary behavior.

We found that sleep restriction induced qualitative changes in food intake, biasing choices toward energy-dense options, without altering total calorie intake. Although some studies have shown increases in calorie intake with sleep deprivation (*Al Khatib et al., 2017*; *Broussard et al., 2016*; *Markwald et al., 2013*; *Patel and Hu, 2008*), our findings are in line with several other

studies (*Cain et al., 2015*; *Nedeltcheva et al., 2009*; *Simon et al., 2015*) and suggest that sleep deprivation induces nuanced changes in food-based decision making, rather than simply increasing hunger or the motivation to eat.

Our results further elaborate on the effects of sleep deprivation on the human brain, suggesting that neural processing of odors is enhanced in primary olfactory brain areas after sleep restriction. Although decoding accuracy can be influenced by factors other than the strength of neural encoding, such as reduced variability (noise), our results indicate that encoding of food vs. non-food odors was more robust in the piriform cortex in a sleep-deprived state. In theory, such enhanced encoding of olfactory information could facilitate odor-evoked approach and consummatory responses (*Aimé et al., 2007*; *Soria-Gómez et al., 2014*). However, we did not observe a direct correlation between encoding in piriform cortex and food intake, and changes in odor information were not directly related to changes in circulating levels of 2-OG. It is possible that nonlinear effects or interactions among different ECS compounds (*Ho et al., 2008*; *Murataeva et al., 2016*) may have obscured a direct linear relationship between 2-OG and encoding in piriform cortex. In any case, our findings indicate that sleep-dependent increases in odor information may not directly mediate the relationship between the ECS and food intake. Instead, they suggest that interactions between olfactory cortex and downstream areas may translate altered chemosensory encoding into changes in food intake.

We found that changes in piriform-insula connectivity were correlated with the effects of sleep deprivation on food choices, suggesting a relationship between sleep-dependent food intake and neural processing in extended olfactory pathways. A formal mediation analysis showed that relative increases in 2-OG were related to reductions in odor-evoked piriform-insula connectivity, which in turn was correlated with increased choices of energy-dense food options. Although these findings need to be confirmed in future studies, they suggest a broader role for neural processing of chemosensory signals along piriform-insula pathways in the regulation of food intake. There are several possible ways by which reduced piriform-insula connectivity could promote choices of energy-dense foods in the presence of enhanced odor information in piriform cortex. Previous studies show that sensory and visceral-homeostatic signals are integrated in the insula, and that this integration is critical for guiding food intake (*de Araujo et al., 2003*; *Livneh et al., 2017*; *Small, 2012*). Reduced piriform-insula connectivity could reflect a diminished integration of chemosensory and homeostatic-visceral information, and a failure to adequately integrate elevated olfactory signals with homeostatic information may drive excess intake of energy-dense food. Alternatively, reduced piriform-insula connectivity could indicate an aberrant assignment of value to energy-dense foods (*Balleine and Dickinson, 2000*; *Gottfried et al., 2003*). Finally, to the degree that sleep deprivation decreases activity in other cortical areas (*Greer et al., 2013*; *Muto et al., 2016*), it is possible that reduced piriform-insula connectivity is related to diminished top-down control over elevated olfactory signals in piriform cortex, which may promote impulsive behavior in response to energy-dense food (*Cedernaes et al., 2014*; *Krause et al., 2017*).

In the current study, we compared brain responses to food odors with high palatability and non-food items with low palatability. As expected, the food odors were rated as higher in pleasantness than non-food odors. In principle, it is therefore possible that our observed effects for food vs. non-food odor encoding in the brain were fully explained by this corresponding pleasantness difference. However, our results remained significant when including pleasantness as a covariate in the statistical models, indicating that in this case pleasantness does not account for our results.

Taken together, our findings show that sleep-dependent changes in food choices are associated with changes in an olfactory pathway that is related to the ECS. This pathway is likely not restricted to sleep-dependent changes in food intake but may also account for dietary decisions more generally. In this regard, our current findings may help to guide the identification of novel targets for treatments of obesity.

## Materials and methods

### Subjects

We consented and screened 41 healthy, right handed, non-smoking, and non-obese men and women between the ages of 18–40 year and body mass index (BMI) between 18.5 and 24.9 kg/m$^2$,

with no history of neurological disorders. We included individuals with a self-reported habitual sleep duration of 7–9 hr, and regular sleep time between 9 pm and midnight. The 7–9 hr habitual sleep inclusion criteria was based on guidelines set by National Sleep Foundation for healthy adults (*Hirshkowitz et al., 2015*). This ensured that all participants had sleep duration and sleep timing that are within the normal range for this age group. A regular sleep time between 9 pm and midnight ensured circadian rhythms were aligned across all participants (*Burgess and Eastman, 2004*), minimizing between-subject variance. Additional exclusion criteria were daytime nap, variable sleep habits, regular night work, travel across time zone during the study, use of medications affecting sleep, and caffeine intake of >300 mg/day. We also administered the Center for Epidemiologic Studies Depression Scale (cut off 16) (*Radloff, 1977*), and the Pittsburgh Sleeping Quality Index (cut off 5) (*Buysse et al., 1989*). Only individuals with scores below the cut off were included in the study. Only non-pregnant women were included.

Of the 41 individuals screened, 29 proceeded with the experimental procedures. Of those, three participants dropped out of the study due to discomfort inside the scanner, and one was excluded from the analysis because of a large number of missed responses. This resulted in a final sample of N = 25 participants (10 male), whose data are reported here. The main outcome measures did not differ between male and female participants (energy density: $T_{23} = 0.618$, p=0.542; 2-OG: $T_{23}=-0.826$, p=0.416; piriform encoding: $T_{23} = 0.095$, p=0.924) and did not correlate with body weight (energy density: $\beta = 0.019$, p=0.865; 2-OG: $\beta=-1.334$, p=0.103; piriform encoding: $\beta = 0.024$, p=0.611). All experimental procedures of this study (STU00203395) were approved by the Institutional Review Board of Northwestern University.

## Experimental procedures overview

In a randomized within-subject crossover protocol, all subjects participated in two sleep sessions that included one night of deprived sleep (DS; 4 hr in bed 1 am – 5 am) and one night of non-deprived sleep (NDS; 8 hr in bed 11 pm – 7 am) at home. There was a washout period of 19 days between the last day of the first session (7 days of sleep stabilization phase, 1 day sleep manipulation, 1 day fMRI session) and the first day of second session (*Figure 1A*). This ensured that the two fMRI days were separated by 28 days for all participants, such that female participants were tested in the same phase of the menstrual cycle. In addition, we recorded self-reported menstrual cycle phase and compared sleep-dependent changes in our primary outcome measures between female participants in the follicular and luteal phase. No significant differences were found between the two menstrual phases (energy-dense food intake: $T_{10}=-0.76$, p=0.465; 2-OG: $T_{10}=-1.41$, p=0.188; piriform encoding: $T_{10}=-0.05$, p=0.960). During the week preceding each session, participants were instructed to maintain a standardized sleep schedule of 7–9 hr sleep (between 10:30 pm and 7:30 am) in order to align the phase of the circadian rhythm across participants. Compliance with the sleep schedule during the sleep stabilization and sleep manipulation phase was monitored using wrist-worn actigraphy and a self-reported sleep diary. Subjects also rated their subjective sleep quality, alertness, and restfulness every morning using an online questionnaire. For the ratings, subjects used a 5-point scale, where one indicated 'poor' or 'least' and five indicated 'excellent' or 'highest'. No naps were allowed during both sleep stabilization and sleep manipulation periods. Participants were also instructed to abstain from alcohol, caffeine, and drugs, including all recreational drugs, to avoid interference with sleep and hormone levels. To ensure that participants limited their caffeine intake to <300 mg/day on sleep stabilization days, they were instructed to consume not more than one small caffeinated drink per day. Foods and drinks high in caffeine (e.g., coffee, chocolate, soda, most tea, including ice tea, traditional black tea, and green tea) were listed in the instruction sheet provided to participants. Subjects were verbally reminded of the caffeine restriction at every study visit. In addition, subjects were instructed to not consume any caffeinated drinks on the scanning day.

On the evening following the night of the sleep manipulation, fMRI scanning was performed after subjects consumed a standardized isocaloric dinner (subjects received exactly the same meal in both sessions). We collected imaging data and blood samples after dinner in the evening following the night of sleep manipulation because previous studies found that experimentally induced sleep deprivation affects ECS compounds and the desire for and consumption of energy-dense food most prominently in this time window (*Hanlon et al., 2016*; *Nedeltcheva et al., 2009*). During fMRI scanning, participants were presented with food odors, non-food odors, and clean air, and rated odor

pleasantness and intensity. Before the fMRI scan in both sessions, participants also rated their subjective sleepiness using the Stanford Sleepiness Scale. To equate food intake leading up to the two fMRI sessions, isocaloric meals were provided for the 24 hr preceding both sessions.

## Sleep protocol and monitoring

Our study used an in-home setting to render the sleep-deprivation protocol as ecologically valid as possible without the additional distractions and stressors of being in an unfamiliar hospital laboratory environment. However, because in-home settings come with potential limitations related to non-compliance, we took several measures to reinforce and monitor compliance with the sleep stabilization and sleep manipulation schedule. For both sleep conditions, participants were instructed to strictly follow the sleep protocol (DS: sleep from 1:00 am to 5:00 am; NDS: Sleep between 11:00 pm and 7:00 am). To encourage compliance with the instructions, research staff discussed strategies to stay awake, such as watching TV, standing up, etc. Sleep and wake-up times were monitored for 8 days (7 days of stabilization, 1 day of manipulation) using a wrist-worn 3-axis accelerometers (Acti-Graph GT9X Link, ActiGraph, LLC, Pensacola, FL) (Ancoli-Israel et al., 2003; Marino et al., 2013). Due to a technical failure, actigraphy data from two participants collected during the sleep stabilization phase of the NDS session was lost (data from both critical nights of sleep manipulation were not affected). Data from one additional participant recorded during one day of the sleep stabilization phase was also lost. Actigraphy data were classified as sleep or awake using Cole-Kripke algorithm, as implemented in the Actilife software. Total sleep time (TST), time in bed (TIB), sleep efficiency (SE), and wake after sleep onset (WASO) were also calculated using the same algorithm. TIB began at sleep onset and ended at awakening, and TST was defined as time sleeping within TIB. WASO was defined as the wake time within TIB, and SE was computed as the ratio between TST and TIB.

In addition to wearing the actigraphy device, subjects also logged their bed time, sleep duration, and sleep quality immediately after scheduled sleep hours in an online sleep diary. Information entered in the sleep diary was time stamped, and study personnel cross-referenced these time stamps and entries with the actigraphy data. Participants were also instructed to avoid daytime naps, and to not consume alcoholic or caffeinated drinks. Participants who failed to follow the sleep protocol were excluded from the study.

## Controlled food intake

During the 24 hr period before each fMRI session, participants were provided with an isocaloric diet. All meals were planned and packaged by a registered dietician at the Clinical Research Unit (CRU) at Northwestern Memorial Hospital, and were based on individually estimated energy requirements according to height, weight, age, and sex. Estimated calorie requirements ranged from 1400 to 2600 kcal/day. Meals were composed of 55–60% carbohydrate, 15–20% protein, and 25–30% fat. On the evening preceding the sleep manipulation, participants arrived at the laboratory and consumed a dinner at 6:00 pm. They were also provided with a take-out breakfast and lunch to be consumed on the following day at 8:00 am and 12:00 pm, respectively. Participants were instructed to not consume any additional foods or drinks, other than water, during the 24 hr period preceding the fMRI session. Breakfast consisted of ~30% of total caloric needs, while the lunch and dinner each consisted of ~35% of the estimated calorie requirement. The standardized pre-scan dinner consisted of an entrée (e.g., hamburger, veggie burger, grilled chicken with dinner roll, ham/turkey sandwich, etc.), a fruit/snack (e.g., apple, banana, granola bar, etc.), and a small non-alcoholic and non-caffeinated drink. The total calorie content of this dinner ranged from 490 to 910 kcal (~35% of estimated calorie requirements) and was composed of 55–60% carbohydrate, 15–20% protein, and 25–30% fat. All participants consumed the entire meal. For each participant, the same meals were provided in both fMRI sessions. All subjects reported that they did not consume any other foods or caffeinated and caloric drinks.

## Collection of blood samples and assay

Upon arrival at the imaging center, an MRI safe catheter was placed in participant's left arm and samples were collected through antecubital venipuncture. Blood plasma samples were collected at baseline (before dinner) and at four time points following the dinner. Blood serum samples for ECS analysis were only collected at 7:30 pm while subjects were inside the MRI scanner, 90 min after the

initiation of dinner (in between the 2nd and 3rd fMRI run). Prior to drawing each sample, 1.5–2 mL of blood were drawn off to remove potentially diluted blood from the dead space of the catheter. Samples were put on ice, centrifuged, aspirated, divided into aliquots, and stored at −80°C until assay.

Serum levels of 2-arachidonoylglycerol (2-AG) and 2-oleoylglycerol (2-OG) were extracted using the Bond Elut C18 solid-phase extraction columns (1 ml; Varian Inc, Lake Forest, CA). Serum samples were processed and the two compounds were quantified using chemical ionization liquid chromatography/mass spectrometry (LC-ESI-MS; Agilent LC-MSD 1100 series, Ramsey, MN), as previously described (*Patel et al., 2005*).

We used enzyme-linked immunosorbent assay (ELISA) kits for total plasma ghrelin (human ghrelin, Millipore), plasma leptin (human leptin, Millipore), and cobas e411 analyzer (Roche) for plasma insulin, and serum cortisol.

## Assessment of hunger

On the day of fMRI scanning, subjects completed both paper and computerized versions of visual analogue scales to measure motivation to eat at baseline (before dinner) and 30 min, 60 min, 90 min, and 120 min after dinner initiation and after the *ad libitum* buffet (see below). At each time-point, sensation of hunger, fullness, satisfaction, and prospective food consumption was assessed with following questions: (1) How hungry do you feel? (2) How full do you feel? (3) How satisfied do you feel? (4) How much do you think you can eat? Ratings were made on a 10 cm visual analogue scale with text at each end indicating the most positive and most negative rating.

## *Ad libitum* buffet and next day food intake

After completion of the fMRI scan, participants were presented with excessive portion sizes of energy-dense sweet (cinnamon roll, donut holes, chocolate chip cookies, mini muffins) and savory food items (hash browns, garlic bread, pizza bites, potato chips) in an all-you-can-eat buffet-style setting. In both sessions participants were instructed to wait in a separate room for 30 min before filling out another questionnaire. This waiting room contained the buffet of food options, and participants were told that they could consume the food freely while they waited, if they wanted. Food items were weighed before and after to determine the amount of food consumed. Total calorie and energy density (kcal/g) of consumed food was calculated from the product nutrition labels. Participants were presented with identical buffet options in both testing sessions.

To track food intake on the day following the experiment (after a night of unrestricted sleep), participants were asked to record their food intake for 24 hr following the fMRI scan using a food diary. To estimate total calories and fat consumed, nutritional information for each consumed food item was obtained from the United States Department of Agriculture (USDA) Food Composition Databases (ndb.nal.usda.gov/ndb/). Percent of calories consumed as fat was calculated by multiplying total grams of consumed fat by nine kcal/g and dividing this number by the total number of calories consumed.

## Odor selection and delivery

During initial screening, participants rated the pleasantness of six energy-dense food odors in randomized order (pot roast, potato chips, garlic bread, cinnamon roll, caramel, and gingerbread, provided by International Flavors and Fragrances [New York City, NY] and Kerry [Tralee, Ireland]). Ratings were made on visual analog scales using a scroll wheel and mouse button press. Anchors were 'most-liked sensation imaginable' (10) and 'most disliked sensation available' (−10). Based on these ratings, two savory and two sweet odors were chosen for each participant such that they were matched in pleasantness, and these odors were used for the remainder of the experiment. After odor selection, participant also rated the pleasantness, intensity (anchors 'strongest sensation imaginable' and 'weakest sensation imaginable'), edibility (anchors 'certainly edible' and 'certainly inedible'), and quality (anchors 'clearly savory' and 'clearly sweet') of the four selected food odors and two non-food control odors (fir needle and celery seed). Odor ratings collected during screening are summarized in *Supplementary file 2*. Most importantly, edibility ratings for food and non-food odors collected during the screening session differed significantly between food and non-food odors ($T_{24}$ = 12.69, p=3.87×10$^{-12}$).

For all odor ratings and the experimental task, odors were delivered directly to subjects' nose using a custom-built MR-compatible olfactometer (*Howard and Kahnt, 2017*; *Howard and Kahnt, 2018*; *Suarez et al., 2019*), capable of redirecting medical grade air with precise timing at a constant flow rate of 3.2 L/min through the headspace of amber bottles containing liquid solutions of the odors. The olfactometer is equipped with two independent mass flow controllers (Alicat, Tucson, AZ), allowing for dilution of odorants with odorless air. At all times throughout the experiment, a constant stream of odorless air is delivered to participants' noses, and odorized air is mixed into this airstream at specific time points, without changing in the overall flow rate. Thus, odor presentation does not involve a change in somatosensory stimulation induced by the airstream.

## Olfactory fMRI task

On the evening following the sleep manipulation, all participants arrived at the scanning center after a 6 hr fast and consumed dinner at 6:00 pm before entering the scanner at 6:45 pm. Immediately before the fMRI scan, participants repeated rated the pleasantness, edibility, and quality of the four selected food odors and the two non-food control odors. Each scanning session consisted of four fMRI runs, and each run consisted of 63 pseudo-randomized trials of olfactory stimulation. On each trial, after a 2 s countdown, the white crosshair in the center of the screen turned blue, cuing participants to sniff the odor for 2.5 s. The sniff cue was followed by a rating scale (pleasantness or intensity, counterbalanced) for 5 s and a 1–8 s inter-trial interval. Each run consisted of nine presentations of the two sweet food odors, the two savory food odors, the two non-food control odors, and clean air (totaling 63 trials/run).

## fMRI data acquisition

Functional MRI data were acquired on a Siemens 3T PRISMA system equipped with a 64-channel head-neck coil. In each scanning run, 382 Echo-Planar Imaging (EPI) volumes were acquired with a parallel imaging sequence with the following parameters: repetition time, 2 s; echo time, 22 ms; matrix size, 104 × 96; field-of-view, 208 × 192 mm (resulting in an in-plane resolution of 2 × 2 mm$^2$); flip angle, 90°; multi-band acceleration factor, 2; slice thickness, 2 mm; 58 slices; no gap; acquisition angle, ~30° rostral to intercommissural line to minimize susceptibility artifacts in piriform cortex (*Deichmann et al., 2003*; *Weiskopf et al., 2006*). *Figure 3—figure supplement 2* shows a normalized group average EPI. A high-resolution (1 mm isotropic) T1-weighted structural scan was also acquired at the beginning of the fMRI session. To support the co-registration of functional and structural images, we also collected 10 whole-brain EPI volumes using the same parameters as the functional EPIs, except for 96 slices and a repetition time of 3.22 s.

## Respiratory data acquisition and analysis

Respiration, as an indirect measure of nasal sniffing, was measured using a MR-compatible breathing belt (BIOPAC Systems Inc, Goleta, CA) affixed around the participant's torso, and recorded at 1 kHz using PowerLab equipment (ADInstruments, Dunedin, New Zealand). Respiratory traces for each fMRI run were temporally smoothed using a moving window of 500 ms, high-pass filtered (cutoff, 50 s) to remove slow-frequency drifts, normalized by subtracting the mean and dividing by the standard deviation across the run trace, and down-sampled to 0.5 Hz for use as nuisance regressors in fMRI data analyses (see below).

For quantification of respiratory peak amplitude and respiratory latency, trial-specific respiratory traces were baseline corrected by subtracting the mean signal across the 0.5 s window preceding sniff cue onset, and then normalized by dividing by the maximum respiratory amplitude of all trials in the run. Respiratory traces were sorted by sleep session and odor, and averaged across trials. Respiratory amplitude was then calculated as the max signal within 5 s of sniff cue onset, and respiratory latency was calculated as the time from sniff cue onset to max amplitude.

## fMRI pre-processing

Pre-processing of fMRI data was performed using SPM12 (http://www.fil.ion.ucl.ac.uk/spm/software/spm12/). For each participant, we aligned all functional volumes for both sleep sessions to the first acquired functional volume to correct for head motion. We then realigned and averaged the ten whole-brain EPI volumes, and co-registered the mean whole-brain EPI to the anatomical T1 image.

The mean functional volume was then co-registered to the mean whole-brain EPI, and this transformation was applied to all functional volumes. Spatial normalization was performed by normalizing the T1 anatomical images to the MNI (Montreal Neurological Institute) space using the six tissue probability map provided by SPM12. For multivariate analysis, the resulting deformation fields were applied to searchlight-based maps of decoding accuracy (see below). The normalized decoding accuracy maps were spatially smoothed with a 6 × 6 × 6 mm full-width half-maximum (FWHM) Gaussian kernel before group-level statistical testing. For functional connectivity and univariate analyses (see below), the motion-corrected and co-registered functional images were normalized to MNI space using the previously estimated deformation fields and spatially smoothed with a 6 × 6 × 6 mm FWHM Gaussian kernel.

To quantify and compare head motion between sessions we computed the average (across scans) of the absolute volume-by-volume displacements for each of the six realignment parameters for each session. We then computed composite scores for translation and rotation parameters and compared them between sessions. There were no sleep-dependent differences in these composite head motion parameters (translation, $T_{24} = 0.25$, p=0.803; rotation, $T_{24}=-0.14$, p=0.891).

## Multivoxel pattern analysis

We implemented a searchlight-based multi-voxel pattern analysis (MVPA) (*Howard and Kahnt, 2018*; *Kahnt et al., 2011*) to decode information about food vs. non-food odors. We first estimated general linear models (GLM) for each subject, separately for each session, using the non-normalized and un-smoothed functional images. The GLM included three regressors of interest specifying onset times for the following conditions: 1) food odors, 2) non-food odors, 3) clean air. We also included the following nuisance regressors: the smoothed and normalized respiratory trace, down-sampled to scanner temporal resolution (0.5 Hz); the six realignment parameters (three translations, three rotations), calculated for each volume during motion correction; the derivate, square, and the square of the derivative of each realignment regressor; the absolute signal difference between even and odd slices, and the variance across slices in each functional volume (to account for fMRI signal fluctuation caused by within-volume head motion); additional regressors as needed to model out individual volumes in which particularly strong head motion occurred (absolute difference between odd and even slices >5 SD or slice variance >4 SD). The parameter estimates from the first two regressors of this GLM reflect the voxel-wise response amplitudes for food and non-food odors, separately for each run and sleep session.

Next, we used these voxel-wise parameter estimates in a searchlight-based, leave-one-run-out cross-validated decoding approach. We decoded food vs. non-food odors from patterns of odor-evoked activity, separately for each of the two sleep sessions. We used The Decoding Toolbox (TDT) to implement the searchlight (*Hebart et al., 2014*) and LIBSVM (*Chang and Lin, 2011*) for the linear support vector machine (SVM) classifier. To test for brain regions that encoded food vs. non-food odors, at each searchlight (sphere with 8 mm radius), we trained a SVM to discriminate between activity patterns evoked by food vs. non-food odors in three of the four runs per session (DS or NDS), and tested it on activity patterns evoked by food vs. non-food from the fourth 'left out' run of the same session. The procedure was repeated four times leaving a different run out, and decoding accuracies were averaged and mapped to the center voxel of the searchlight. This procedure was repeated for every voxel within a 10% gray-matter mask (based on SPMs tissue probability map that was inverse-normalized into the individual native space, as described in *Howard and Kahnt, 2018*). The resulting accuracy maps for food vs. non-food odors for DS and NDS sessions were subtracted (DS >NDS), normalized, and smoothed (6 mm FWHM). We tested for significant differences between DS and NDS sessions at the group level using voxel-wise one-sample t-tests. Statistical thresholds were set to p<0.05, family-wise error (FWE) small-volume corrected for multiple comparisons at the voxel-level in a functional mask of piriform cortex that was obtained from a one-sample t-test of decoding accuracy for food vs. non-food odors, averaged across sleep sessions (p<0.001, see *Figure 3—figure supplement 1*).

## Functional connectivity analysis

We used the generalized psycho-physiological interaction (PPI) model (*McLaren et al., 2012*) to test for regions in which sleep-dependent changes in functional connectivity with the piriform cortex

correlated with sleep-dependent changes in food intake. The seed region in the piriform cortex was defined from significant voxels (p<0.001) in the contrast DS >NDS for decoding food vs. non-food odors. We first specified session-wise (DS or NDS) PPI models at the single-subject level using normalized and smoothed functional images. Odor presentation (odor vs. clean air) was used as 'psychological variable' and mean piriform cortex activity as 'physiological variable'. The PPI models also included the same nuisance regressors as described above for the GLM for the MVPA analysis. Estimated connectivity parameters for odor vs. no-odor were contrasted between DS and NDS sessions, and entered into a group-level model with changes in energy-dense food intake as regressor of interest. We tested for regions in which sleep-dependent changes in odor-evoked functional connectivity (odor >clean air) correlated significantly with sleep-dependent changes in food intake. Statistical thresholds were set to p<0.05, FWE small-volume corrected for multiple comparisons at the voxel-level in an anatomical mask of insula cortex obtained using the Automated Anatomical Labeling (AAL) atlas.

## Univariate fMRI analysis

We conducted a univariate analysis using the traditional GLM approach on normalized and spatially smoothed functional images. The session-wise GLM included three regressors of interest specifying onsets for the following conditions: 1) food odors, 2) non-food odors, 3) clean air. The GLM included the same nuisance regressors as the GLM described in the MVPA section. To test for odor-evoked fMRI activity in the piriform cortex, contrast images for food and non-food odors > clean air trials were created at the single-subject level and averaged across sessions. Group-level analyses were carried out using voxel-wise one-sample t-tests thresholded at p<0.05, FWE whole-brain corrected. To test for sleep-dependent differences in odor-evoked activity in the piriform cortex, we extracted parameter estimates for the three odor conditions per sleep session from a piriform cortex region of interest (defined using the odor >clean air contrast, at p<0.001, see *Figure 4*). We computed a two-way ANOVA on the parameter estimates to test for sleep-dependent main effects and interactions.

## Acknowledgements

The authors thank Dr. CJ Hillard and team for MS analysis of serum samples, International Flavor and Fragrances (A Dumer and RS. Santos) and Kerry (JL Buckley) for providing food odorants. This work was supported by the Comprehensive Metabolic Core at Northwestern University, the National Center for Advancing Translational Sciences, Grant number UL1 TR001422 (to TK), the National Blood Lung and Heart Institute grant T32 HL007909 (to SB), the National Institute of Diabetes and Digestive and Kidney Diseases grant R21 DK118503 (to TK), and the National Institute on Deafness and Other Communication Disorders grant R01 DC015426 (to TK).

## Additional information

### Competing interests

Thorsten Kahnt: Reviewing editor, *eLife*. The other authors declare that no competing interests exist.

### Funding

| Funder | Grant reference number | Author |
| --- | --- | --- |
| National Institute of Diabetes and Digestive and Kidney Diseases | R21 DK118503 | Thorsten Kahnt |
| National Institute on Deafness and Other Communication Disorders | R01 DC015426 | Thorsten Kahnt |
| National Center for Advancing Translational Sciences | UL1 TR001422 | Thorsten Kahnt |
| National Heart, Lung, and Blood Institute | T32 HL007909 | Surabhi Bhutani |

The funders had no role in study design, data collection and interpretation, or the decision to submit the work for publication.

## Author contributions
Surabhi Bhutani, Conceptualization, Formal analysis, Investigation, Writing—original draft; James D Howard, Formal analysis, Investigation, Writing—review and editing; Rachel Reynolds, Investigation, Writing—review and editing; Phyllis C Zee, Jay Gottfried, Conceptualization, Writing—review and editing; Thorsten Kahnt, Conceptualization, Formal analysis, Supervision, Funding acquisition, Writing—original draft

## Author ORCIDs
James D Howard ![ORCID] https://orcid.org/0000-0002-9309-3773
Thorsten Kahnt ![ORCID] https://orcid.org/0000-0002-3575-2670

## Ethics
Human subjects: Informed consent was obtained from all subjects and all experimental procedures of this project (STU00203395) were approved by the Institutional Review Board of Northwestern University.

## Decision letter and Author response
Decision letter https://doi.org/10.7554/eLife.49053.048
Author response https://doi.org/10.7554/eLife.49053.049

## Additional files
### Supplementary files
• Supplementary file 1. Sleep monitoring (actigraphy) data. Means ± SEM of time in bed (TIB), total sleep time (TST), wake after sleep onset (WASO), and sleep efficiency (SE) in the DS and NDS session for the week of sleep stabilization (average across seven nights) and the night of sleep manipulation. P-values from paired t-tests of the difference between NDS and DS sessions.
DOI: https://doi.org/10.7554/eLife.49053.044

• Supplementary file 2. Ratings of odor stimuli during screening session. Means ± SEM of ratings for pleasantness, intensity, quality (sweet vs savory), and edibility for the six food and two non-food odors used in this study. For each subject, four food and two non-food odors were presented. Numbers in parenthesis indicate the number of subjects for which a given odor was selected. F- and P-values from one-way ANOVAs across odors.
DOI: https://doi.org/10.7554/eLife.49053.045

• Transparent reporting form
DOI: https://doi.org/10.7554/eLife.49053.046

### Data availability
All data generated or analysed during this study are included in the manuscript, supporting files and source data files.

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
