## [Decision Letter]

Thank you for submitting your article "Olfactory contributions to sleep-dependent food intake in humans" for consideration by *eLife*. Your article has been reviewed by three peer reviewers, including, including X as the Reviewing Editor and Reviewer #1 as the Reviewing Editor and Reviewer #1, and the evaluation has been overseen by Christian Büchel as the Senior Editor. The following individuals involved in review of your submission have agreed to reveal their identity: Noam Sobel (Reviewer #2).

The reviewers have discussed the reviews with one another and the Reviewing Editor has drafted this decision to help you prepare a revised submission.

Summary:

This study examines how a manipulation of sleep duration impacts both food choice and circulating endocannabinoid system (ECS) compounds, as well as how this association is mediated by altered connectivity of neural olfactory regions. Consistent with previous work they found that sleep restriction increases circulating levels of an ECS compound (2-OG) and increases food intake (measured as energy density in food selected from a buffet). Using an MVPA approach, the authors identified regions that were selective in their representations of food vs. non-food smells, with the right piriform cortex exhibiting a group difference in representational distances of food vs. non-food odors. Using this as a seed region, the authors found that individual differences in odor vs. clean air task connectivity between the piriform cortex and insula mediated the individual differences observed in ECS compounds and food choice.

All three reviewers felt that this is potentially a very important study that addresses a very important question, with immediate interest for both scientists and the general public alike. Moreover, it is clearly the product of "high level" rigorous science. More specifically, the authors tackle their question from several directions at once. This is on one side a clear advantage, yet on the other, when you do a lot of different work, there are lots of potential complications that need to be addressed before the manuscript can be accepted.

Essential revisions:

1) Imaging analysis.

Reviewer 1 pointed out that the critical finding that sets this work apart from prior studies is the observation that task-related piriform-insula connectivity differences statistically mediate a relationship between 2-OG and food choice. Yet a lot relies on the veracity of the neuroimaging analysis. Both estimates of task effects (e.g., GLM) and connectivity measures in the BOLD response are highly sensitive to motion and physiological artifacts that can sometimes lead to spurious observations. This brings up several concerns.

First, reviewer 1 is concerned that the food > non-food difference in piriform representations may be due to differences in head motion across groups. It seems possible that sleep manipulations can lead to differences in head motion in the scanner, that in turn would alter the reliability of the representational distance estimates that the SVN decoder picks up on (leading to a potential spurious group difference). The authors should both report whether there were differences in framewise displacement measures across groups and include a displacement measure as a nuisance variable in the estimation of the group differences.

Second, reviewer 1 pointed out that while there is no statistical difference in sniff amplitude (Figure 1—figure supplement 3E), there does appear to be a mean difference such that the sleep restricted group is taking lower amplitude sniffs. Even if this isn't significant at the behavioral level, it might lead to differences in the expression of respiratory artifacts in the BOLD response. In addition, there may be differences in respiration variability, both during the odor presentation and in-between, which is also known to contribute to artifacts in the BOLD signal (see Birn et al., (2008). Therefore, the authors should include models of respiration artifacts, particularly in the connectivity estimates, in order to rule out possible spurious associations due to group differences in respiration. This concern was echoed by reviewer 3, who pointed out that, they were not convinced that there were not, for instance, motoric differences in the sniff that contribute to the effects. Although Figure 1—figure supplement 3E and F show a non-significant differences, when one cuts out C and D and overlay them up against the light, it sure appears that the odors for NDS are equal to the clean air in DS. In other words, it is not clear where the amplitude and latency measures come from, but they don't seem to capture the differences you can see if you overlap the curves in C and D. One other concern in this regard is that if the piriform-insula connectivity measure is for odors (collapsed) v. clean air, then this sniff response comparison should be for odors (collapsed) v. clean air. (One note on food odors v. non-food odors: it's confusing to me that celery seed is considered a non-food odor as this is a not-uncommon ingredient in food dishes.)

Finally, all three reviewers raised concerns about the use of different contrasts used in the analyses. Reviewer 1 points out that the search for group differences in representational distances in the piriform uses the food > non-food contrast, but then the connectivity results between piriform and insula are using the odor > clean air contrast. It is likely that this is because the food > non-food contrast did not produce a significant result. This should still be reported and clarified in the text.

2) Pleasantness confounds.

Both reviewers 1 and 2 had concerns regarding the role of subjective pleasantness preferences had on the task. Reviewer 2 pointed out that a major analysis in this manuscript is the searchlight-based multi-voxel pattern analysis contrasting the response to "food" and "non-food" odors. As clearly indicated in Figure 1—figure supplement 3, "food" odors were much (as in p = 2.5x10-6) more pleasant than "non-food" odors. Thus, why is this to be considered a contrast of "food vs. non-food" and not a contrast of "pleasant vs. less-pleasant". The impact of sleep deprivation on hedonics is also interesting, but not the aim of this manuscript. This potential confound is so blatant, that the reader must be missing something. Thus, what is missing? Why is the sharp difference in pleasantness between the edible and non-edible odorants not a concern? The authors can do one of two things: Either better explain how this relates directly to food vs. non-food odors, or run analyses to address this. For example, either regress out pleasantness differences, or select subsets of data devoid of this potential confound.

Related to this, reviewer 2 pointed out that the selection of non-food odorants was slightly odd. After all, celery seed is not that far from a food. Why didn't the authors just use perfume? It would have addressed their pleasantness difference, and it's clearly not edible. Odorant selection, however, is behind us. Thus, what the authors should at least add is the actual edibility ratings of the odorants. This should be provided in a supplementary table with the associated statistics of the differences in edibility.

3) Validity of inferences.

Reviewers 2 and 3 had concerns about the certainty of the conclusions being made. Reviewer 2 pointed out that the authors conclude that "sleep deprivation induces changes in food intake through the modulation of an olfactory pathway that is related to the endocannabinoid system". This very strong statement implies causation, that it is not 100% certain the authors have in hand. There is a relationship, but the authors to make such a strong causal claim, wouldn't they need to somehow find a way to independently manipulate the olfactory pathway, and show that it is indeed responsible for the effect? Would sleep deprivation fail to impact eating behaviour of individuals with anosmia? This is a major concern, but it is of course trivial to address: The authors should either slightly tone down the claims on causation (the manuscript is strong enough as it is) or make a better substantiated claim on causation.

Reviewer 3 pointed out that the conclusions of the paper emphasize a finding that olfaction contributes to sleep-dependent food intake in humans (see Title). This is an overstatement of the results. First, unlike other papers, the present study did not find a difference in food intake (caloric intake) based on sleep deprivation. The authors did find a difference in food preference (food decisions). Second, the measure of olfaction in the result is not olfaction per se but rather piriform-insula connectivity when sleep deprived v. not sleep deprived. This should be made more clear and up front throughout the manuscript.

4) 2-OG vs. food choice association.

All three reviewers had a concern about the association between 2-OG and food choice. An inspection of Figure 2B suggests that the association between ECS system and food choice may be driven by 5 participants. Reviewers 2 and 3 pointed out that if these outliers are removed, the association appears to become negative. The authors seem to be aware of this since they used a robust regression analysis; however, robust regression approaches simply account for differences in variance of each observation in the overall estimate. Reviewer 1 recommend a non-parametric statistic, such as a bootstrap or permutation test. (Note: This is listed as a Major concern because this is one of the critical observations for the authors' primary conclusions).

5) Compliance.

Reviewer 3 raised concerns about subject compliance. There is a lot of trust in participants. Trust that they did not consume additional food and drinks, trust they did not take caffeine, and trust in both the participant and actigraph that the sleep restriction instructions were complied with. Given the importance of these, it is surprising that participants were not kept in-lab for the sleep deprivation manipulation. The authors should address this.

6) Sniffing vs. inhalation.

Reviewer 2 raised a concern about the nature of the inhalation measure identified as sniffing. This reviewer points out that authors don't directly measure sniffing; they measure respiration with respiratory belts. However, the path from thoracic and abdominal movements to airflow patterns in the nose is complex, and more critically, it is variable. In fact, even in direction; an expanding abdomen can reflect inhalation and it can reflect exhalation. Moreover, belts don't discriminate between nasal and oral inhalation. What if a participant didn't like a particular odorant, and shifted to oral inhalation every time it was presented? How would the authors know this?

Moreover, as the authors know, sniffing is odorant dependent. Thus, the pleasant food odors may have been greeted with slightly more vigorous sniffs, yet the less pleasant non-food odors with slightly less vigorous sniffs. If this difference was very small, it would not be picked up by the belts. Such a small difference, however, may also be very consistent, and thus meaningful. All this is critical, because the authors may attribute an activation pattern to a difference in odor, where in fact it may reflect a difference in sniffing. Alas, these are indeed often very difficult to untangle, but this still needs to be addressed. This is especially true in light of the simplicity of precisely measuring sniffing at the nose. If the olfactometer uses a mask, then a simple pressure tube off the mask accurately converts sniffing. If no mask is used, then a nasal cannula provides an even better measure of sniffing. As sniffing is very easy to measure, it is not clear why the authors did not do so.

As to the current manuscript, the authors should use more careful terminology. In Figure 1—figure supplement 3, the authors should refer to respiratory amplitude, not sniff amplitude. Reviewer 2 points out that this is a problem with the entire field, but should be reasonably addressed here. In future, reviewer 2 suggests the authors precisely measure sniffing in their studies. It's cheap, easy, and informative.

---

## [Author Response]

Summary:This study examines how a manipulation of sleep duration impacts both food choice and circulating endocannabinoid system (ECS) compounds, as well as how this association is mediated by altered connectivity of neural olfactory regions. […] More specifically, the authors tackle their question from several directions at once. This is on one side a clear advantage, yet on the other, when you do a lot of "things", there are lots of potential complications that need to be addressed before the manuscript can be accepted.

We thank the reviewers for their positive and thoughtful comments, and the reviewing editor for preparing the summary and the consolidated comments. Below we outline how we have addressed the essential revisions in the revised manuscript.

Essential revisions:1) Imaging analysis.Reviewer 1 pointed out that the critical finding that sets this work apart from prior studies is the observation that task-related piriform-insula connectivity differences statistically mediate a relationship between 2-OG and food choice. Yet a lot relies on the veracity of the neuroimaging analysis. Both estimates of task effects (e.g., GLM) and connectivity measures in the BOLD response are highly sensitive to motion and physiological artifacts that can sometimes lead to spurious observations. This brings up several concerns.First, reviewer 1 is concerned that the food > non-food difference in piriform representations may be due to differences in head motion across groups. It seems possible that sleep manipulations can lead to differences in head motion in the scanner, that in turn would alter the reliability of the representational distance estimates that the SVN decoder picks up on (leading to a potential spurious group difference). The authors should both report whether there were differences in framewise displacement measures across groups and include a displacement measure as a nuisance variable in the estimation of the group differences.

We thank the reviewers for these suggestions. We were also particularly concerned about potential differences in head motion between the two sleep conditions. We therefore included 24 (instead of the standard 6) motion regressors along with nuisance regressors that capture within-volume motion (absolute difference between odd and even slices, variance across slices) into all GLMs.

We also directly compared head motion between the two sessions (i.e., NDS and DS). Specifically, we computed the average (across scans) of the absolute volume-by-volume displacements for each of the 6 realignment parameters for each session. We found no significant sleep-dependent differences in any of these parameters (x, P=0.476; y, P=0.962; z, P=0.922; pitch, P=0.879; roll, P=0.9182; yaw, P=0.894). In addition, we computed composite scores for translation and rotation parameters, which showed no sleep-dependent differences (translation, T_24_=0.25, P=0.803; rotation, T_24_=−0.14, P=0.891).

Finally, as suggested, we added these two composite scores as covariates in the second level models. Adding these covariates did not change the results for the decoding (x=20 y=8 z=−12, T_23_=6.12, P_FWE-SVC_=0.0001) or the connectivity analysis (right insula, x=40, y=6, z=0, T_21_=5.42, P_FWE-SVC_=0.021).

We have added the results of these control analyses to the manuscript.

Materials and methods section:

“To quantify and compare head motion between sessions we computed the average (across scans) of the absolute volume-by-volume displacements for each of the 6 realignment parameters for each session. We then computed composite scores for translation and rotation parameters and compared them between sessions. There were no sleep-dependent differences in these composite head motion parameters (translation, T_24_=0.25, P=0.803; rotation, T_24_=−0.14, P=0.891).”

Legend Figure 3:

“This result did not change when including covariates for head motion (translation and rotation) into the group-level model (x=20 y=8 z=−12, T_23_=6.12, P_FWE-SVC_=0.0001).”

Legend Figure 5:

“This result did not change when including covariates for head motion (translation and rotation) into the group-level model (x=40, y=6, z=0, T_21_=5.42, P_FWE-SVC_=0.021).”

Second, reviewer 1 pointed out that while there is no statistical difference in sniff amplitude (Figure 1—figure supplement 3E), there does appear to be a mean difference such that the sleep restricted group is taking lower amplitude sniffs. […] One other concern in this regard is that if the piriform-insula connectivity measure is for odors (collapsed) v. clean air, then this sniff response comparison should be for odors (collapsed) v. clean air. (One note on food odors v. non-food odors: it's confusing to me that celery seed is considered a non-food odor as this is a not-uncommon ingredient in food dishes.)

The reviewers point out a number of important issues. Similar to head motion, we controlled for potential respiratory confounds by including respiratory traces as covariates in all GLMs used for the decoding and connectivity analyses. We also tested whether the respiratory measures differed between sessions, the results of which are reported in the legend accompanying Figure 1—figure supplement 3.

In addition to these controls, we have computed new GLMs which include respiratory nuisance regressors that model the “respiration response function” as defined in Birn et al., (2008). Controlling for breathing using these more advanced nuisance regressors did not change the results of the decoding (x=20, y=8, z=−12, T_24_=4.83, P_FWE-SVC_=0.001) or connectivity analyses (x=40, y=6, z=0, T_23_=5.59, P_FWE-SVC_=0.011). Taken together, these new control analyses suggest that our findings are unlikely to be driven by respiratory artifacts.

Regarding sleep-dependent differences in odor-evoked respiratory amplitude and latency, we do not fully understand the reviewer’s concern when they write “it is not clear where the amplitude and latency measures come from but they don't seem to capture the differences you can see if you overlap the curves in C and D”. We fully agree that the peak of the respiratory trace for food in NDS (light pink trace in Figure 1—figure supplement 3C) appears to roughly line up with the peak of the respiratory trace for clean air in DS (dark gray trace in Figure 1—figure supplement 3D). We would like to point out that these respiratory traces (averaged across trials and subjects) are presented in panels C and D only for visualization, and the across-subjects mean values of respiratory peak amplitude and latency directly corresponding to these traces are plotted in panels E and F of the same figure using bar plots with error bars. We suspect that the confusion may have originated from the fact that the respiratory traces themselves in Figure 1—figure supplement 3C and D did not contain error bars, which are necessary to interpret the mean differences. We now include error bars on these traces to better illustrate that differences between respiratory responses to different odors are within what can be expected by chance.

Finally, as requested, we also computed statistical tests for comparing odors (collapsed) vs. clean air trials which show that there are no sleep-dependent effects in respiratory peak amplitude or latency (sleep-by-odor ANOVA on amplitude; main effect sleep F_1,24_=1.45, P=0.240; main effect odor F_1,24_=22.41, P<0.0001; sleep-by-odor interaction, F_1,24_=1.24, P=0.277; sleep-by-odor ANOVA on latency; main effect sleep F_1,24_=0.14, P=0.715; main effect odor F_1,24_=3.50, P=0.074; sleep-by-odor interaction, F_1,24_=0.53, P=0.474).

We have added the results of these additional analyses to the manuscript.

Legend Figure 3:

“Finally, controlling for respiratory response functions (Birn et al., 2008) did not change the result (x=20, y=8, z=−12, T_24_=4.83, P_FWE-SVC_=0.001).”

Legend Figure 5:

“In addition, controlling for respiratory response functions (Birn et al., 2008) did not change the result (x=40, y=6, z=0, T_23_=5.59, P_FWE-SVC_=0.011).”

Legend Figure 1—figure supplement 3:

“There were also sleep-depended effects on amplitude and latency when comparing odor (collapsed across food and non-food odors) vs. clean air trials (sleep-by-odor ANOVA on amplitude; main effect sleep F_1,24_=1.45, P=0.240; main effect odor F_1,24_=22.41, P<0.0001; sleep-by-odor interaction, F_1,24_=1.24, P=0.277; sleep-by-odor ANOVA on latency; main effect sleep F_1,24_=0.14, P=0.715; main effect odor F_1,24_=3.50, P=0.074; sleep-by-odor interaction, F_1,24_=0.53, P=0. 474).”

We address the question about the choice of celery seed as a non-food odor below.

Finally, all three reviewers raised concerns about the use of different contrasts used in the analyses. Reviewer 1 points out that the search for group differences in representational distances in the piriform uses the food > non-food contrast, but then the connectivity results between piriform and insula are using the odor > clean air contrast. It is likely that this is because the food > non-food contrast did not produce a significant result. This should still be reported and clarified in the text.

We used the food vs. non-food contrast in our decoding analysis to test for differences in encoding of odor information. For the connectivity analysis, we tested odor-evoked connectivity, thus contrasting connectivity during odor vs. clean air trials. Reviewers are correct that the contrast between food and non-food odor for sleep-dependent changes in connectivity did not reveal significant effects. We now report this in the manuscript.

Results section:

“In contrast, sleep-dependent changes in piriform connectivity for food vs. non-food odors did not show a significant relationship with changes in food choices.”

2) Pleasantness confounds.Both reviewers 1 and 2 had concerns regarding the role of subjective pleasantness preferences had on the task. Reviewer 2 pointed out that a major analysis in this manuscript is the searchlight-based multi-voxel pattern analysis contrasting the response to "food" and "non-food" odors. As clearly indicated in Figure 1—figure supplement 3, "food" odors were much (as in p = 2.5x10-6) more pleasant than "non-food" odors. Thus, why is this to be considered a contrast of "food vs. non-food" and not a contrast of "pleasant vs. less-pleasant". The impact of sleep deprivation on hedonics is also interesting, but not the aim of this manuscript. This potential confound is so blatant, that the reader must be missing something. Thus, what is missing? Why is the sharp difference in pleasantness between the edible and non-edible odorants not a concern? The authors can do one of two things: Either better explain how this relates directly to food vs. non-food odors, or run analyses to address this. For example, either regress out pleasantness differences, or select subsets of data devoid of this potential confound.

The objective of this experiment was to study sleep-dependent brain responses to food odors corresponding to highly palatable energy-dense food items (caramel, cinnamon bun, potato chips, pot roast, garlic bread, etc.), relative to odors of objects (in principle edible) with low palatability and low energy density (celery seed and fir needle). We expected that higher palatability would be reflected in higher pleasantness ratings, and thus view the pleasantness difference as validation of our experimental design with respect to stimulus selection.

However, we understand the reviewers’ concerns and conducted additional analyses to test whether our effects are merely driven by pleasantness. In these analyses, we controlled for pleasantness in the single-subject GLMs used to estimate activity patterns evoked by food and non-food odors by adding the ratings as a nuisance regressor. Interestingly, despite the fact that food and non-food odors differed in pleasantness, we still found a significant (but much weaker) effect in our piriform ROI, such that odor encoding was enhanced in the sleep-deprived session (x=20, y=8, z=−10, T_24_=3.12, P_FWE-SVC_=0.045).

In addition, it is worth emphasizing that pleasantness ratings for food and non-food odors did not differ between the two sleep sessions, and that controlling for individual differences in the pleasantness of food vs. non-food odors in the group-level analysis did not change our decoding results (x=20, y=8, z=−12, T_23_=6.34, P_FWE-SVC_=0.0001).

We now include these results into the manuscript and mention the potential pleasantness confound in the Discussion section.

Legend Figure 3:

“In addition, results were still significant when including covariates for odor pleasantness in the first- (x=20, y=8, z=−10, T_24_=3.12, P_FWE-SVC_=0.045) and group-level models (x=20, y=8, z=−12, T_23_=6.34, P_FWE-SVC_=0.0001).”

Discussion section:

“In the current study, we compared brain responses to food odors with high palatability and non-food items with low palatability. As expected, the food odors were rated as higher in pleasantness than non-food odors. In principle, it is therefore possible that our observed effects for food vs. non-food odor encoding in the brain were fully explained by this corresponding pleasantness difference. However, our results remained significant when including pleasantness as a covariate in the statistical models, indicating that in this case pleasantness does not account for our results.”

Related to this, reviewer 2 pointed out that the selection of non-food odorants was slightly odd. After all, celery seed is not that far from a food. Why didn't the authors just use perfume? It would have addressed their pleasantness difference, and it's clearly not edible. Odorant selection, however, is behind us. Thus, what the authors should at least add is the actual edibility ratings of the odorants. This should be provided in a supplementary table with the associated statistics of the differences in edibility.

Edibility ratings did indeed differ significantly between food and non-food odors (T_24_=12.69, P=3.87x10^-12^). We have now included a new supplementary table (Supplementary file 2) summarizing the odor ratings (collected during the screening session) for all odors used in the study.

Materials and methods section:

“Odor ratings collected during screening are summarized in Supplementary file 2. Most importantly, edibility ratings for food and non-food odors collected during the screening session differed significantly between food and non-food odors (T_24_=12.69, P=3.87x10^-12^).”

3) Validity of inferences.Reviewers 2 and 3 had concerns about the certainty of the conclusions being made. Reviewer 2 pointed out that the authors conclude that "sleep deprivation induces changes in food intake through the modulation of an olfactory pathway that is related to the endocannabinoid system". This very strong statement implies causation, that it is not 100% certain the authors have in hand. There is a relationship, but the authors to make such a strong causal claim, wouldn't they need to somehow find a way to independently manipulate the olfactory pathway, and show that it is indeed responsible for the effect? Would sleep deprivation fail to impact eating behaviour of individuals with anosmia? This is a major concern, but it is of course trivial to address: The authors should either slightly tone down the claims on causation (the manuscript is strong enough as it is) or make a better substantiated claim on causation.

We have toned down our conclusions regarding causation throughout the manuscript. In particular, we have changed the sentence referenced above to:

Discussion section:

“Taken together, our findings show that sleep-dependent changes in food choices are associated with changes in an olfactory pathway that is related to the ECS.”

Reviewer 3 pointed out that the conclusions of the paper emphasize a finding that olfaction contributes to sleep-dependent food intake in humans (see Title). This is an overstatement of the results. First, unlike other papers, the present study did not find a difference in food intake (caloric intake) based on sleep deprivation. The authors did find a difference in food preference (food decisions). Second, the measure of olfaction in the result is not olfaction per se but rather piriform-insula connectivity when sleep deprived v. not sleep deprived. This should be made more clear and up front throughout the manuscript.

The reviewer is correct that the relationship between energy-dense food choices and total calorie intake is not evident in our study. We further agree that energy-dense food choice was correlated with connectivity of the olfactory cortex, not olfaction per se.

We have changed wording throughout the manuscript and also changed the Title to:

“Olfactory connectivity mediates sleep-dependent food choices in humans”

4) 2-OG vs. food choice association.All three reviewers had a concern about the association between 2-OG and food choice. An inspection of Figure 2B suggests that the association between ECS system and food choice may be driven by 5 participants. Reviewers 2 and 3 pointed out that if these outliers are removed, the association appears to become negative. The authors seem to be aware of this since they used a robust regression analysis; however, robust regression approaches simply account for differences in variance of each observation in the overall estimate. Reviewer 1 recommend a non-parametric statistic, such as a bootstrap or permutation test. (Note: This is listed as a Major concern because this is one of the critical observations for the authors' primary conclusions).

Following this suggestion, we computed a permutation test to verify the result from the robust regression analysis. This permutation test (100,000 permutations) confirmed that the correlation between 2-OG and food choice is statistically significant (p=0.018).

We have added this result to the manuscript.

Results section:

“Interestingly, sleep-dependent increases in 2-OG correlated significantly with increases in the energy density of food consumed at the post-scanning buffet (robust regression, β=0.47, P=0.027; permutation test, P=0.018; Figure 2B).”

5) Compliance.Reviewer 3 raised concerns about subject compliance. There is a lot of trust in participants. Trust that they did not consume additional food and drinks, trust they did not take caffeine, and trust in both the participant and actigraph that the sleep restriction instructions were complied with. Given the importance of these, it is surprising that participants were not kept in-lab for the sleep deprivation manipulation. The authors should address this.

We agree that the issue of compliance is an inherent concern in sleep and dietary manipulation studies conducted in the field. We opted for an in-home sleep manipulation in order to render the sleep deprivation as ecologically valid as possible without the typical distractors and stressors of being in an unfamiliar hospital laboratory environment. However, we took several measures to monitor compliance and to control for potential confounds. Most importantly, we used wrist actigraphy to monitor subjects’ sleep-wake schedule, as reported in Figure 1—figure supplement 1. We also collected a time stamped self-reported sleep diary and ratings of sleep quality during both sessions (Figure 1D and E) to verify that our sleep protocol was effective.

Importantly, non-compliance with the sleep instructions would have presumably increased between subject variability, thereby decreasing the likelihood of observing significant effects. Thus, assuming that some subjects may have “cheated”, the resulting observations are a conservative estimate of the true effect of sleep deprivation on the brain activity and behavior reported here.

With regards to food intake, two of the four meals provided to the participants prior to scanning were consumed at the laboratory and imaging center, respectively. We also provided subjects with repeated instructions to not consume any additional foods or snacks in the 24 hours preceding the scanning session. Regarding the remaining two take-out meals, subjects were required to report whether they consumed all of the provided food, and they were also asked to report any additional foods or drinks consumed.

We now discuss these issues in the manuscript.

Materials and methods section:

“Our study used an in-home setting to render the sleep-deprivation protocol as ecologically valid as possible without the additional distractions and stressors of being in an unfamiliar hospital laboratory environment. However, because in-home settings come with potential limitations related to non-compliance, we took several measures to reinforce and monitor compliance with the sleep stabilization and sleep manipulation schedule.”

6) Sniffing vs. inhalation.Reviewer 2 raised a concern about the nature of the inhalation measure identified as sniffing. […] In future, reviewer 2 suggests the authors precisely measure sniffing in their studies. It's cheap, easy, and informative.

We appreciate the methodological advice from reviewer 2. We now refer to “respiration” not “sniff” in Figure 1—figure supplement 3 and throughout the manuscript. We also now clearly state in the methods that respiratory effort bands only provide an indirect measure of sniffing.

Materials and methods section:

“Respiration, as an indirect measure of nasal sniffing, was measured using a MR-compatible breathing belt (BIOPAC Systems Inc, Goleta, CA) affixed around the participant’s torso, and recorded at 1 kHz using PowerLab equipment (ADInstruments, Dunedin, New Zealand).”